# Societal Trust Related to COVID-19 Vaccination: Evidence from Western Balkans

**Smiljana Cvjetkovic [1,2], Vida Jeremic Stojkovic [1,2], Stefan Mandic-Rajcevic [2,3], Sanja Matovic-Miljanovic [2,*], Janko Jankovic [3], Aleksandra Jovic Vranes [3], Aleksandar Stevanovic [3] and Zeljka Stamenkovic [3]**

[1] Department of Humanities, Faculty of Medicine, University of Belgrade, 11000 Belgrade, Serbia
[2] Euro Health Group, A/S (Denmark), Regional Office in Serbia, 11000 Belgrade, Serbia
[3] Institute of Social Medicine, Faculty of Medicine, University of Belgrade, 11000 Belgrade, Serbia
[*] Correspondence: smatovic@ehg.dk; Tel.: +381-63-1014848

**Abstract:** The lower rates of COVID-19 vaccination in Western Balkans countries could be partially explained by societal distrust of its citizens, jeopardizing the sustainability of COVID-19 vaccination programs. The aim of the study was to determine the level and predictors of societal trust in five countries of the region. Using an online questionnaire, data were obtained from 1157 respondents from Albania, Bosnia and Herzegovina, Montenegro, North Macedonia, and Serbia. The instrument included a socio-demographic questionnaire, a measure of vaccination behavior, and a scale measuring societal trust. Being a significant determinant of the COVID-19 vaccination behavior in all countries, societal trust considerably varied from country to country (F (24, 4002) = 7.574, $p < 0.001$). It was highest in North Macedonia (Mean = 3.74, SD = 0.99), and lowest in Albania (Mean = 3.21, SD = 1.03). Younger, female, less religious, and higher educated tended to have more pronounced societal trust in Serbia. In North Macedonia, younger age and lower health literacy predicted societal trust, while in Bosnia and Herzegovina, educational level was the single predictor. In Montenegro and Albania, higher societal trust was significantly predicted by lower health literacy only. The results provide evidence that the determinants of societal trust in Western Balkans vary across countries, indicating the need for different approaches in communication campaigns.

**Keywords:** societal trust; Western Balkans; COVID-19 vaccination

## 1. Introduction

Societal trust, as a relationship that exists between individuals and/or a system in which one party accepts a vulnerable position, assuming the competence of the other, in exchange for a reduction in decision complexity [1], is increasingly important in the domain of public health, especially in the time of health emergencies such as the COVID-19 pandemic. Societal trust can be directed both vertically, towards authorities and institutions such as government, politicians, health authorities, science, etc. (institutional trust), and horizontally, towards fellow citizens (interpersonal trust) [2].

The level of institutional trust is an important determinant of the overall success of public health interventions [3]. Empirical evidence shows that trust in government and scientific institutions increases the population's compliance with public health policies and guidelines [4,5]. In the context of vaccination, individual citizens accept vulnerable positions in relation to the public health authority/government and choose to trust them to help make a decision on whether to get vaccinated or not [1]. When it comes to the COVID-19 vaccination, in many countries worldwide, there is a significant number of those who choose not to receive the vaccine, and there is not a small number of those who have a pronounced vaccine hesitancy [6–9]. Hence, instead of explaining vaccine hesitancy (as a state of uncertainty and indecision about vaccination) [10] only by misunderstanding and insufficient information among the public, contemporary accounts suggest that vaccine hesitancy also reflects poor societal trust in medical and scientific institutions [11]. Research

has shown that trust in government and medical authorities is positively associated with vaccine uptake [1]. In addition, results of a large international survey show that the strongest correlates of willingness to get COVID-19 vaccines are trust in medical doctors and scientists [12].

Societal trust varies greatly between countries. In higher-income countries, institutional trust is higher compared to low-income countries. Additionally, institutional trust is higher in countries with authoritarian governments compared to democratic states [13], but this finding should be taken with caution because they do not necessarily reflect the real picture of public opinions due to self-censorship, which is not a rare phenomenon in authoritarian societies [14].

Institutional trust in countries of the Western Balkans has been particularly low for a long period [15]. Results of a recent large-scale opinion poll show that distrust in government has been exacerbated in the Western Balkans countries during the times of the pandemic [16] and suggest a direct association between the distrust in government and vaccine hesitancy of their citizens. Significant COVID-19 vaccine hesitancy in the population of Western Balkans is reflected in low vaccine coverage despite the availability of vaccines in some of the countries. The shares of the population that has completed the initial COVID-19 vaccination protocol in the Western Balkan countries are below 50%, which significantly lags behind the EU average, which is 73.2% of the population receiving the initial protocol [17]. Having in mind this large discrepancy in COVID-19 vaccination rates, it should be emphasized that COVID-19 vaccination programs could not be sustainable at the national, regional, and global levels if every country does not achieve sufficient vaccination coverage [18]. Exploring various factors affecting vaccine acceptance is a critical step toward this goal.

Early adults are considered to be the social group bearing a large burden of confinement measures and social distancing [19]. Although they have a low risk of developing serious COVID-19 disease compared to the other age groups, young adults are identified as important agents of COVID-19 transmission, particularly because they are often asymptomatic. In addition, emerging evidence from longitudinal studies suggests that mental health consequences of the COVID-19 pandemic have been observed in adolescents, youth, and young adults [20]. Therefore, their compliance with anti-epidemic measures, including vaccination, is of critical importance for many reasons.

In order to design communication campaigns to satisfy the information needs of those who distrust institutions, it is important to identify the precise agents that they distrust. Therefore, the aim of our study was to investigate the level of different types of societal trust (trust in government, medical institutions, scientists, pharmaceutical companies, and family physicians) having a particular focus on COVID-19 vaccination, and to determine predictors of societal trust in early adults in five Western Balkans countries (Albania, Bosnia and Herzegovina, Montenegro, North Macedonia, and Serbia). Additionally, we compared the level of societal trust between countries and explored the association between the level of societal trust and the vaccination status of respondents.

## 2. Materials and Methods

### 2.1. Sampling and Procedure

The present study is part of a larger project examining COVID-19-related vaccine hesitancy in Western Balkans. A cross-sectional study was carried out in the period from July to October 2021 and included adult citizens aged above 18 from the five Western Balkans countries: Albania, Bosnia and Herzegovina, North Macedonia, Montenegro, and Serbia. Convenience sampling was applied, and data were collected by online questionnaire, disseminated using the SurveyMonkey platform, which automatically stores digital responses to a database. The questionnaire was shared through widely used online social media (Facebook, Instagram), using a targeted posting method. The duration needed to complete the questionnaire was 15–20 min. Individuals could access the questionnaire link by clicking on the post, and 1605 of them fully completed the questionnaire. For the

purpose of the study, subsample of the early adult participants aged 18–45 was selected (in total, 1157 respondents). We defined early adulthood as an age period ranging from 18 to 45 years, based on Levinson's theory of individual life structure [21,22].

*2.2. Measures*

The comprehensive instrument was developed for the purpose of this study based on the literature [23–25]. It included:

(1)   The short scale measuring social trust consisting of 6 items on a five-point agreement scale (ranging from 1 "strongly disagree" to 5 "strongly agree"). Items represent trust in political authorities (one item), health authorities (one item), family physician (one item), pharmaceutical industry (two items), and scientists (one item). Factor analysis confirmed the one-factor structure of the scale. The Cronbach's alpha for six items ranged from 0.87 in the Albanian sample to 0.91 in the Serbian sample, which indicated high internal consistency. The total score was calculated by summing the responses to all items and dividing that sum by the number of items (six). For the purpose of calculating the total score, the responses on items 1, 2, and 6 were reversely coded. The total score range was divided into four quartiles: 1–1.99 (highly negative), 2–2.99 (moderately negative), 3–3.99 (moderately positive), and 4–5 (highly positive).

(2)   Socio-demographic characteristics included eight items: gender, age, education level, employment status, financial status, marital status, having children, and religiousness.

(3)   Health-related characteristics included three items:

-   The existence of chronic health conditions was assessed by the single question, "Do you suffer from any chronic disease?" with the binary (Yes/No) response;
-   General health status was assessed with the 5-point Likert scale ranging from "Very good" to "Very bad";
-   Had COVID-19—with the binary (Yes/No) response.

(4)   COVID-19 vaccination behavior was evaluated by the single item measure—"Have you been vaccinated?" question with the binary (Yes/No) response.

(5)   European Health Literacy Assessment Questionnaire [26] contains a total of 16 questions divided into three areas of inquiry: (1) Healthcare (7 questions), (2) Disease Prevention (5 questions), and (3) Health promotion (4 questions). Answers are given through a four-point Likert scale (1 = very difficult, 2 = difficult, 3 = easy, 4 = very easy). The indices for health literacy were standardized to unified metrics from 0 to 50 using the formula; Index = (Mean − 1) × (50/3), resulting in Health Literacy Index (HLI). HLI was categorized into four levels: Inadequate (0–25), Problematic (>25–33), Sufficient (>33–42), and Excellent (>42–50).

*2.3. Statistical Analysis*

Descriptive statistics were used to detail the sample characteristics and to summarize the variables. Internal consistency of the scale scores was evaluated with Cronbach's alpha. The associations between societal trust level and COVID-19 vaccination behavior in each country were assessed by means of logistic regression. To examine the significant differences between countries in the overall level of societal trust, analyses of variance (ANOVA) with Tuckey's test for post hoc comparisons were used. In addition, a multivariate analysis of variance (MANOVA) was conducted utilizing individual items of the Societal trust scale as dependent variables, with country as independent. Bonferroni correction was used to adjust for multiple tests. To identify determinants of societal trust in each country, linear regressions were conducted. Variables found to be significant in univariate analysis were entered in multiple analysis for each country. All analyses were performed in Statistical Package for Social Sciences (SPSS) for Windows, version 25 (IBM Corp., Armonk, NY, USA), and $p < 0.05$, namely $p < 0.008$, was considered statistically significant.

*2.4. Ethical Considerations*

The survey was anonymous, collecting no personally identifiable information from participants. The participants were informed about the purpose of the study during the introductory portion of the survey, and they provided their consent by ticking the specified box. Participants were free to stop responding to the survey at any point. Participants received no incentives for their participation to ensure voluntariness.

## 3. Results

*3.1. Socio-Demographic and Health-Related Characteristics*

The total number of respondents in the study was 1157, of which 224 were from Albania, 179 from Bosnia and Herzegovina, 158 from Montenegro, 231 from North Macedonia, and 365 from Serbia. The socio-demographic and health-related characteristics of the respondents are presented in Table 1. The proportion of women ranged from 65.5% in the Serbian sample to 77.8% in the Montenegrin sample. Educational level differed among countries: in Albania, the largest proportion had a bachelor's degree (43.3%), similarly to North Macedonia (45.7%), Montenegro (45.6%), and Serbia (39.2%), while in Bosnia and Herzegovina majority had only high school (62.0%). Employment status also varied largely among countries: while in Serbia, only 21.9% of respondents were unemployed, in Bosnia and Herzegovina, 57.5% reported being unemployed, similarly to 51.9% in North Macedonia. The majority of respondents in all countries declared themselves religious.

**Table 1.** Sample description.

| Variables | Albania | Bosnia and Herzegovina | Montenegro | North Macedonia | Serbia |
|---|---|---|---|---|---|
| Gender | | | | | |
| Male | 71 (31.7%) | 57 (31.8%) | 35 (22.2%) | 57 (24.7%) | 126 (34.5%) |
| Female | 153 (68.3%) | 122 (68.2%) | 123 (77.8%) | 174 (75.3%) | 239 (65.5%) |
| Education level | | | | | |
| High school | 20 (8.9%) | 111 (62.0%) | 65 (41.1%) | 99 (43.0%) | 97 (26.6%) |
| Bachelor's degree | 97 (43.3%) | 58 (32.4%) | 72 (45.6%) | 105 (45.7%) | 143 (39.2%) |
| Master's degree | 92 (41.1%) | 10 (5.6%) | 15 (9.5%) | 22 (9.6%) | 91 (24.9%) |
| Ph.D. | 15 (6.7%) | 0 (0.0%) | 6 (3.8%) | 4 (1.7%) | 34 (9.3%) |
| Employment status | | | | | |
| Employed | 130 (58.0%) | 65 (36.3%) | 82 (51.9%) | 98 (42.4%) | 240 (65.8%) |
| Self-employed | 23 (10.3%) | 11 (6.1%) | 19 (12.0%) | 13 (5.6%) | 45 (12.3%) |
| Unemployed | 71 (31.7%) | 103 (57.5%) | 57 (36.1%) | 120 (51.9%) | 80 (21.9%) |
| Material status | | | | | |
| Very good | 24 (10.7%) | 31 (17.3%) | 14 (8.9%) | 19 (18.2%) | 47 (12.9%) |
| Good | 63 (28.1%) | 79 (44.1%) | 58 (36.7%) | 77 (33.3%) | 137 (37.5%) |
| Average | 118 (52.7%) | 64 (35.8%) | 77 (48.7%) | 111 (48.1%) | 154 (42.2%) |
| Bad | 18 (8.0%) | 4 (2.2%) | 7 (4.4%) | 20 (8.7%) | 25 (6.8%) |
| Very bad | 1 (0.4%) | 1 (0.6%) | 2 (1.3%) | 4 (1.7%) | 2 (0.5%) |
| Marital status | | | | | |
| Single | 111 (49.6%) | 130 (72.6%) | 74 (46.8%) | 174 (75.3%) | 151 (41.1%) |
| Married | 87 (38.8%) | 43 (24.0%) | 60 (38.0%) | 49 (21.2%) | 157 (43.0%) |
| Cohabitation | 24 (10.7%) | 1 (0.6%) | 18 (11.4%) | 8 (3.5%) | 48 (13.2%) |
| Divorced | 1 (0.4%) | 4 (2.2%) | 6 (3.8%) | 0 (0.0%) | 7 (1.9%) |
| Widowed | 1 (0.4%) | 1 (0.6%) | 0 (0.0%) | 0 (0.0%) | 2 (0.5%) |
| Having children | | | | | |
| Yes | 84 (37.5%) | 44 (24.6%) | 69 (43.7%) | 42 (18.2%) | 163 (44.7%) |
| No | 140 (62.5%) | 135 (75.4%) | 89 (56.3%) | 189 (81.8%) | 202 (55.3%) |

**Table 1.** *Cont.*

| Variables | Albania | Bosnia and Herzegovina | Montenegro | North Macedonia | Serbia |
|---|---|---|---|---|---|
| Religiousness | | | | | |
| Yes | 145 (64.7%) | 142 (79.3%) | 111 (70.3%) | 155 (67.1%) | 218 (59.7%) |
| No | 79 (35.3%) | 37 (20.7%) | 47 (29.7%) | 76 (32.9%) | 147 (40.3%) |
| General health status | | | | | |
| Very good | 102 (45.5%) | 67 (37.4%) | 51 (32.2%) | 88 (38.1%) | 99 (27.1%) |
| Good | 97 (43.3%) | 86 (48.0%) | 81 (51.3%) | 104 (45.0%) | 194 (53.2%) |
| Average | 22 (9.8%) | 25 (14%) | 24 (15.2%) | 34 (14.7%) | 65 (17.8%) |
| Bad | 3 (1.3%) | 1 (0.6%) | 2 (1.3%) | 5 (2.2%) | 7 (1.9%) |
| Very bad | 0 (0.0%) | 0 (0.0%) | 0 (0.0%) | 0 (0.0%) | 0 (0.0%) |
| Chronic disease | | | | | |
| Yes | 25 (11.2%) | 22 (12.3%) | 31 (19.6%) | 30 (13.0%) | 69 (18.9%) |
| No | 199 (88.8%) | 157 (87.7%) | 127 (80.4%) | 201 (87.0%) | 296 (81.1%) |
| Had COVID-19 | | | | | |
| Yes | 118 (52.7%) | 45 (25.1%) | 69 (43.7%) | 87 (37.7%) | 123 (33.7%) |
| No | 106 (47.3%) | 134 (74.9%) | 89 (56.3%) | 144 (62.3%) | 242 (66.3%) |
| Health Literacy Index * | | | | | |
| Inadequate | 135 (82.3%) | 8 (5.2%) | 107 (76.4%) | 131 (71.6%) | 19 (6.5%) |
| Problematic | 17 (10.4%) | 43 (27.7%) | 23 (16.4%) | 27 (14.8%) | 94 (32.2%) |
| Sufficient | 7 (4.3%) | 74 (47.7%) | 3 (2.1%) | 17 (9.3%) | 117 (40.1%) |
| Excellent | 5 (3.0%) | 30 (19.4%) | 7 (5.0%) | 8 (4,4%) | 62 (21.0%) |

* Questions of the Health Literacy Survey were answered by 990 respondents due to sample shedding.

The majority of respondents in Serbia, Bosnia, and Herzegovina, Montenegro, and North Macedonia had not had COVID-19 infection (ranging from 56.3% to 74.9%), while in Albania, 52.7% reported having had COVID-19. Over 80% of respondents in all countries reported not having any pre-existing chronic health condition. Around half of the respondents in each sample assessed their general health status as good.

### 3.2. Societal Trust and COVID-19 Vaccination Behavior

The level of societal trust in all studied countries was moderately positive. There was a statistically significant difference in the level of total societal trust related to COVID-19 epidemics between the respondents living in studied countries, $F(24, 4002) = 7.574$, $p < 0.001$; Wilk's $\lambda = 0.856$, partial $\eta^2 = 0.0$. Highest level of total societal trust related to the epidemics was observed in North Macedonia (Mean = 3.74, SD = 0.99), followed by Montenegro (Mean = 3.62, SD = 1.09), Serbia (Mean = 3.40, SD = 1.13) and Bosnia and Herzegovina (Mean = 3.23, SD = 1.04), while the lowest level was in Albania (Mean = 3.21, SD = 1.03).

Regarding scores on particular items of the scale, results are presented in Table 2. Statistically significant differences were present among countries for the six items of the Societal trust scale.

Post-hoc tests revealed that respondents from Albania and Bosnia and Herzegovina were more convinced that vaccination against COVID-19 is largely promoted by pharmaceutical companies in order to gain financial profits compared to respondents from Serbia ($p < 0.008$) and North Macedonia ($p < 0.008$). In accordance with the previous, in North Macedonia, compared to Bosnia and Herzegovina ($p < 0.008$) and Albania ($p < 0.008$), respondents were significantly less inclined to believe that pharmaceutical companies are reluctant to publish comprehensive and detailed research reports on the risks of adverse reactions to vaccines. Additionally, respondents from Serbia were less susceptible to this belief compared to the respondents from Albania ($p < 0.008$).

**Table 2.** Comparison between participants from different countries regarding six items of the social trust scale.

| | Country | N | M | SD | df | F | *p* | Partial η² |
|---|---|---|---|---|---|---|---|---|
| Vaccination against COVID-19 is largely promoted by pharmaceutical companies in order to gain financial profits. | Albania | 224 | 3.05 | 1.40 | 4 | 10.612 | <0.008 | 0.04 |
| | Bosnia and Herzegovina | 179 | 3.03 | 1.37 | | | | |
| | Montenegro | 158 | 2.65 | 1.43 | | | | |
| | North Macedonia | 231 | 2.40 | 1.37 | | | | |
| | Serbia | 365 | 2.50 | 1.45 | | | | |
| Pharmaceutical companies are reluctant to publish comprehensive and detailed research reports on the risks of adverse reactions to vaccines. | Albania | 224 | 3.38 | 1.37 | 4 | 7.932 | <0.008 | 0.03 |
| | Bosnia and Herzegovina | 179 | 3.31 | 1.33 | | | | |
| | Montenegro | 158 | 3.00 | 1.41 | | | | |
| | North Macedonia | 231 | 2.77 | 1.42 | | | | |
| | Serbia | 365 | 2.89 | 1.49 | | | | |
| I believe that health authorities when they encourage vaccination, do so with the best intentions. | Albania | 224 | 3.60 | 1.35 | 4 | 12.867 | <0.008 | 0.04 |
| | Bosnia and Herzegovina | 179 | 3.46 | 1.30 | | | | |
| | Montenegro | 158 | 4.03 | 1.31 | | | | |
| | North Macedonia | 231 | 3.92 | 1.28 | | | | |
| | Serbia | 365 | 3.28 | 1.45 | | | | |
| I believe that political authorities when they encourage vaccination, do so with the best of intentions. | Albania | 224 | 3.12 | 1.47 | 4 | 15.066 | <0.008 | 0.05 |
| | Bosnia and Herzegovina | 179 | 3.25 | 1.33 | | | | |
| | Montenegro | 158 | 3.84 | 1.41 | | | | |
| | North Macedonia | 231 | 3.70 | 1.40 | | | | |
| | Serbia | 365 | 3.00 | 1.49 | | | | |
| Family physicians have an important role in educating people about the importance of vaccination against COVID-19. | Albania | 224 | 3.99 | 1.25 | 4 | 3.252 | 0.012 | 0.01 |
| | Bosnia and Herzegovina | 179 | 3.83 | 1.27 | | | | |
| | Montenegro | 158 | 3.97 | 1.36 | | | | |
| | North Macedonia | 231 | 4.23 | 1.19 | | | | |
| | Serbia | 365 | 3.88 | 1.33 | | | | |
| I think that the principal motive for the scientists who participated in the creation of the vaccine against COVID-19 was profit. | Albania | 224 | 3.00 | 1.46 | 4 | 13.015 | <0.008 | 0.04 |
| | Bosnia and Herzegovina | 179 | 2.78 | 1.31 | | | | |
| | Montenegro | 158 | 2.49 | 1.36 | | | | |
| | North Macedonia | 231 | 2.22 | 1.22 | | | | |
| | Serbia | 365 | 2.36 | 1.35 | | | | |

The conviction that the health authorities, when they encourage vaccination, do so with the best intentions was the most pronounced among the participants in Montenegro, where it was significantly higher compared to Serbia (*p* < 0.008) and Bosnia and Herzegovina (*p* < 0.008). In addition, this belief was significantly stronger in North Macedonia compared to Serbia (*p* < 0.008) and Bosnia and Herzegovina (*p* < 0.008).

The lowest level of trust in political authorities was noted in Serbia, followed by Albania, where participants in both countries were significantly less inclined to believe that political authorities recommend vaccines with the best intentions compared to Montenegro (*p* < 0.008) and North Macedonia (*p* < 0.008). At the same time, this kind of trust was significantly lower in Bosnia and Herzegovina compared to Montenegro (*p* < 0.008).

The attitude that the principal motive for the scientists who participated in the creation of the vaccine against COVID-19 was profit was the most strongly pronounced in Albania, and it was significantly higher than in Serbia (*p* < 0.008), Montenegro (*p* < 0.008) and North Macedonia (*p* < 0.008). Additionally, people in Bosnia and Herzegovina were more inclined to this belief compared to people in Serbia (*p* < 0.008) and North Macedonia (*p* < 0.008).

Societal trust score was a significant predictor of COVID-19 vaccination behavior in all countries (Table 3). The association was the strongest in Serbia, followed by North Macedonia.

**Table 3.** Societal trust as a predictor of COVID-19 vaccination status in different countries.

|  | Albania | Bosnia and Herzegovina | Montenegro | North Macedonia | Serbia |
|---|---|---|---|---|---|
|  | OR (95% CI) | OR (95% CI) | OR (95% CI) | OR (95% CI) | OR (95% CI) |
| Societal trust | 2.28 (1.64–3.18) ** | 2.83 (1.87–4.28) ** | 2.18 (1.55–3.07) ** | 3.40 (2.37–4.89) ** | 5.15 (3.69–7.17) ** |

Notes: ** $p < 0.01$.

### 3.3. Determinants of Societal Trust

Determinants of societal trust in different countries are presented in Table 4.

**Table 4.** Determinants of societal trust in different countries (standardized coefficients).

|  | Albania | Bosnia and Herzegovina [a] | Montenegro | North Macedonia [b] | Serbia [c] |
|---|---|---|---|---|---|
| Constant | 3.626 ** | 3.485 ** | 4.068 ** | 4.925 ** | 4.515 ** |
| Age |  |  |  | −0.16 * | −0.13 * |
| Gender<br>Female<br>Male |  | Ref.<br>−0.14 |  |  | Ref.<br>−0.14 * |
| Education<br>High school<br>Bachelor's degree<br>Master's degree<br>Doctoral degree |  | Ref.<br>0.15 *<br>0.21 ** |  |  | Ref.<br>0.01<br>0.20 **<br>0.18 ** |
| Religiousness<br>No<br>Yes |  |  |  |  | Ref.<br>−0.15 ** |
| HLI | −0.24 ** |  | −0.23 ** | −0.30 ** | 0.08 |
| R | 0.24 | 0.28 | 0.23 | 0.33 | 0.35 |
| $R^2$ (Adjusted) | 0.05 | 0.07 | 0.05 | 0.10 | 0.11 |

Notes: * $p < 0.05$; ** $p < 0.01$; HLI-Health Literacy Index; a—MLR adjusted for gender and education; b—MLR adjusted for age and HLS; c—MLR adjusted for age, gender, education, and religiousness.

In the Serbian sample, all variables, with the exception of health literacy, added statistically significantly to the prediction ($p < 0.05$). Younger respondents holding master's and doctoral degrees tended to have more pronounced societal trust related to COVID-19 vaccination, while males and religious had less societal trust (F (7.316) = 6.417, $p < 0.001$, $R^2 = 0.124$)).

In the sample collected in Bosnia and Herzegovina only education statistically significantly predicted societal trust (F (3.175) = 5.102, $p < 0.01$, $R^2 = 0.080$)). Participants holding bachelor's and master's degrees expressed to a greater extent the societal trust related to COVID-19 vaccination compared to their counterparts holding high school degrees.

In the sample gained from North Macedonia, the younger and less health literate were significantly more likely to manifest societal trust related to COVID-19 vaccination (F (2.185) = 11.432, $p < 0.01$, $R^2 = 0.110$)).

The health literacy statistically significantly predicted societal trust in Montenegro (F (1.140) = 8.138, $p < 0.01$, $R^2 = 0.060$)), and Albania as well (F (1.165) = 10.280, $p < 0.01$, $R^2 = 0.060$)). In both countries, respondents with a lower level of health literacy were more likely to have a higher level of societal trust related to COVID-19 vaccination.

## 4. Discussion

To our knowledge, this is the first study examining societal trust related to COVID-19 vaccination in the population of early adults in the Western Balkans. A relatively high level

of societal trust in respondents of all studied countries is promising; however, significant differences existed between countries in the total score and in scores on particular items of societal trust, as well. The country with the highest observed level of total social trust was North Macedonia, while the lowest level was noted in Bosnia and Herzegovina and Albania. The highest level of trust in the total sample was expressed toward family physicians and health authorities, while respondents were least trustful towards pharmaceutical companies. Young adults in Albania and Bosnia and Herzegovina were significantly more distrustful of pharmaceutical companies and scientists. Respondents in Montenegro and North Macedonia had more trust than others in health authorities, while respondents in Serbia and Albania were least convinced that health authorities have the best intentions when recommending vaccination against COVID-19. The lowest level of trust in political authorities was noted in Serbia and Albania.

Similar to our findings, a study of the trust in the health system of Western Balkans' citizens [27] showed that citizens of North Macedonia trusted the health system the most, while citizens of Serbia and Bosnia and Herzegovina the least. These differences among countries point to the possible peculiarities in the national health systems that require attention and analysis in order to increase trust.

According to the Gallup Poll from 2019, citizens from the Western Balkans demonstrated less trust in their national government compared to OECD and OECD-EU countries [28]. In contrast to our findings, in this study, the citizens of Serbia had the highest confidence in the government, while citizens in Bosnia and Herzegovina demonstrated the lowest level of trust towards the national government, which was explained by the associated highest perceived level of corruption by respondents. A similar finding was demonstrated in the survey conducted by the Balkans in Europe Policy Advisory Group (BiEPAG) [16], where respondents in Bosnia and Herzegovina expressed the lowest trust in political institutions, with only 2% of the sample trusting the government. Additionally, similarly to our findings, BiEPAG study results show that doctors and medical staff won the highest trust of citizens in all countries of Western Balkans, suggesting that the healthcare profession still enjoys people's greatest trust. This implies that health authorities should have a pivotal role, together with physicians in primary health, in promoting vaccination and educating the general public in the Western Balkans.

Our study showed inconsistent findings regarding determinants of social trust among five countries of the Western Balkans. Gender, age, and religiousness were significant predictors of trust in the Serbian sample only. Respondents from Serbia who assessed themselves as more religious demonstrated less trust towards societal factors, which is in line with the results of other studies exploring the effect of religiosity on social trust [29]. Our finding that early adults in Serbia expressed higher societal trust is consistent with the results of another study conducted among citizens from the Western Balkans [27], as well as with the results of a recent study of younger adults in Australia [30]. However, not all empirical findings align with ours. For example, Gallup Poll results from 2019 suggested that the most distrustful towards government were citizens of the Western Balkans region aged 15–29 years [28]. In a study conducted in Australia [31], results showed that trust in health and political authorities was significantly higher in older respondents. Similarly, according to the study by Kanellopoulou et al., the trust in the Greek government and Greek health authorities was stronger in older participants [32]. These contradicting results suggest that distrust in institutions can be prevalent in any age group, so strategies to increase trust should be aimed at the population as a whole.

Higher education was a significant determinant of societal trust in Serbia and Bosnia and Herzegovina. This finding aligns with the results of two studies from Australia [30,31], with lower education being associated with lower institutional trust. Additionally, in a study of trust in the healthcare system in Western Balkans, lower trust in the healthcare system was predicted by lower educational level [27]. However, regarding health literacy, in North Macedonia, Albania, and Montenegro, low levels of health literacy predicted higher societal trust in respondents, which is in contrast with the results from recent studies [30,33].

This may be explained by increased criticism in individuals who become knowledgeable and literate due to the access to diverse online information and start to question health authorities, especially at a younger age [34,35]. In addition, the overabundance of information during the COVID-19 pandemic makes it difficult for individuals to differentiate between accurate and false information, causing suspicions and distrust [36]. Evidence shows that people of the Western Balkans are susceptible to conspiracy theories, with nearly 80% of respondents believing in at least one COVID-19-related conspiracy theory [37]. Interestingly, according to this survey, a stronger inclination to conspiracy theories in people of Western Balkans was not associated with lower education level, as is the case in most other countries. Therefore, making linear associations between literacy/education and trust is no longer possible without taking into account the complexities of infodemics in the concrete socio-cultural context.

Finally, the results of our study confirmed the conclusions of previous research that societal trust is a significant predictor of vaccination behavior [12]. In this study, higher total score on the societal trust scale was significantly associated with vaccination acceptance in all studied countries, with the strongest association in respondents from Serbia, followed by respondents in North Macedonia. Similarly, in the other study from the Balkans, trust in the government was a strong predictor of vaccination against COVID-19, especially in the sample from Serbia. Therefore, in order to enhance the sustainability of COVID-19 vaccination programs, efforts to increase societal trust in Western Balkans should be made together with vaccination promotion campaigns. Special attention should be given to the control of misinformation [38,39] and the debunking of conspiracy theories as a global public health concern [40].

Although this is among the first studies exploring societal trust related to COVID-19 vaccination behavior in Western Balkans, several limitations should be briefly stressed. First, we employed convenience sampling, and the survey was administered using online platforms, which could both be sources of unrepresentativeness. Second, the cross-sectional design of the study does not allow conclusions about a causal relationship between variables. In spite of mentioned limitations, the main results of our study suggest that determinants of societal trust in Western Balkans vary across countries, indicating the need for different approaches in communication campaigns with the aim of promoting trust and securing sustainable efforts to control pandemics.

**Author Contributions:** Conceptualization, S.C., V.J.S. and S.M.-M.; Methodology, S.C., V.J.S., S.M.-R. and J.J.; Software, S.M.-R.; Validation, S.C. and S.M.-R.; Formal Analysis, S.C.; Investigation, S.C., V.J.S., S.M.-R., J.J., A.J.V., A.S., Z.S. and S.M.-M.; Resources, S.M.-M.; Data Curation, S.C. and S.M.-R.; Writing—Original Draft Preparation, S.C. and V.J.S.; Writing—Review and Editing, S.M.-M., S.M.-R., J.J., A.J.V., A.S. and Z.S.; Visualization, S.C.; Supervision, S.M.-M., S.C. and V.J.S.; Project Administration, S.M.-M., S.C. and V.J.S.; Funding Acquisition, S.M.-M. All authors have read and agreed to the published version of the manuscript.

**Funding:** This research was funded by Euro Health Group, A/S (EHG), project number 2474; the APC was funded by Euro Health Group, A/S (EHG).

**Institutional Review Board Statement:** The study was conducted in accordance with the Declaration of Helsinki and approved by The Ethics Commission of the Faculty of Medicine, the University of Belgrade (approval number: 1322/VII-11).

**Informed Consent Statement:** Informed consent was obtained from all subjects involved in the study.

**Data Availability Statement:** Data are available on request from the corresponding author.

**Acknowledgments:** We would like to express our great gratitude to Vesna Bjegović-Mikanović for her valuable help, insightful advice, and suggestions.

**Conflicts of Interest:** The authors declare no conflict of interest.

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
