# Peer review of "Societal Trust Related to COVID-19 Vaccination: Evidence from Western Balkans"

_sustainability, doi:10.3390/su142013547_

Round 1

Reviewer 1 Report

The paper is a quite interesting report on Western Balkans situation regarding societal trust to Covid vaccination. It is well structured and it can add intersting information on the issue also for the intarnational community. 

However, I have some observations on specific points.

Lines 45-46 I'm not sure that misinformation does not represent a key point in vaccine hesitancy (see among others, D'errico et al,  Vaccine). Please modify the sentence and better explain what Authors mean in this affirmation. 

Lines 52-54.  "Societal trust varies greatly between countries. In higher income countries institutional trust is higher compared to low-income countries. Also, institutional trust is higher in countries with authoritarian governments compared to democratic states [13]" It is a very interesting affirmaation that deserves a deeper insight as tha doubt may be that in authoritarian countries institutional trust could be forced. Authors should dwell on this point. 

Lines 69-70. Reference 18 does not refere to a scientific paper. Since the issue is interesting please amplify the scientific background of this point. 

Lines 85-89. In my opinion they are more pertinent to the material and methods section; please remove from introduction section.

Author Response

Dear reviewer,

Thank you very much for your time, insightful comments, recommendations and encouragement to revise and resubmit the manuscript. The manuscript has been revised according to your requirements and suggestions as follows.

On behalf of all the authors,

Vida Jeremic Stojkovic

Response to comments:

Q1: Lines 45-46 I'm not sure that misinformation does not represent a key point in vaccine hesitancy (see among others, D'errico et al,  Vaccine). Please modify the sentence and better explain what Authors mean in this affirmation.

RE1: We agree that misinformation presents one of the key factors of vaccine hesitancy, so we changed this sentence in accordance:

“Hence, instead of explaining vaccine hesitancy (as a state of uncertainty and indecision about vaccination) [10], only by misunderstanding and insufficient information among the public, contemporary accounts suggest that vaccine hesitancy also reflects poor societal trust in medical and scientific institutions[11].”

Q2: Lines 52-54.  "Societal trust varies greatly between countries. In higher income countries institutional trust is higher compared to low-income countries. Also, institutional trust is higher in countries with authoritarian governments compared to democratic states [13]" It is a very interesting affirmaation that deserves a deeper insight as tha doubt may be that in authoritarian countries institutional trust could be forced. Authors should dwell on this point.

RE2: Thank you for this suggestion, you are absolutely right, so we included this potential explanation in the text:

“Also, institutional trust is higher in countries with authoritarian governments compared to democratic states[13], but this finding should be taken with caution because they do not necessarily reflect the real picture of public opinions due to self-censorship, which is not a rare phenomenon in authoritarian societies [1]"

Q3: Lines 69-70. Reference 18 does not refere to a scientific paper. Since the issue is interesting please amplify the scientific background of this point.

RE3: Besides similar polls we couldn’t find any strictly scientific study to support this statement, so we decided to leave this statement out of the manuscript.

Q4: Lines 85-89. In my opinion they are more pertinent to the material and methods section; please remove from introduction section.

RE4: We agree, so we have moved it to the Methods section, lines 105-107.

[1] Maleki, Ammar. How do leading methods mislead? Measuring public opinions in authoritarian contexts. In: IPSA 2021-26th World Congress of Political Science. 2021.

Reviewer 2 Report

General Comments:

The study addresses an important topic in determining the factors that may influence adherence to COVID vaccination in 5 countries from western balkans.

Introduction:

The theoretical framework seems adequate.

Materials and Methods:

The sample and method of access to it are defined, the measurement instruments are explained as well as their operation, validity and reliability values. Statistical methods and respect for ethical issues are explained.

Results:

They should be improved, in some cases they are confusing and the comments are not easily observable in the tables, namely tables 3 and 4 which are not clear, in some cases they do not present OR, CI and p-values.

In my opinion the authors should review and redo this section.

Discussion:

The discussion seems appropriate, but I cannot comment without the respective clarification of the results.

The study has severe limitations with regard to external validity, the data comes from an online survey that may not represent the opinion of the general population, this aspect must be safeguarded.

Author Response

Dear reviewer,

Thank you very much for your time, insightful comments, recommendations and encouragement to revise and resubmit the manuscript. The manuscript has been revised according to your requirements and suggestions as follows.

On behalf of all the authors,

Vida Jeremic Stojkovic

Response to comments:

Q1: Results: They should be improved, in some cases they are confusing and the comments are not easily observable in the tables, namely tables 3 and 4 which are not clear, in some cases they do not present OR, CI and p-values. In my opinion the authors should review and redo this section.

RE1: Table 3 contains the results of logistic regression analysis, while the Table 4 contains the linear regression analysis applied in order to reveal statistically significant predictors, as we previously described in the Method section (Lines 144-145 and 150-152). In order to make the results more clear we added the note under the tables.

Q2: Discussion: The discussion seems appropriate, but I cannot comment without the respective clarification of the results.

RE2: We hope that our previous answer clarified the description of the results.

Q3: The study has severe limitations with regard to external validity, the data comes from an online survey that may not represent the opinion of the general population, this aspect must be safeguarded.

RE3: You are absolutely right, we explicated this limitation in the last paragraph of the manuscript (lines 341-342):

“First, we employed convenience sampling, and the survey was administered using online platforms, which could both be sources of unrepresentativeness.”

Reviewer 3 Report

This paper reports on the reasons citizens choose not to be vaccinated against COVID-19 in four Balkan countries. It is an important issue and having greater representation by countries outside of the US and Europe is critical to address the ongoing coronavirus threat and other threats to come.  Despite the need for English editing, the paper is understandable and easy to read.  The sample size seems adequate to address the questions asked and the only concern is the potential selection bias.

Comments and suggestions:

1.  The frequency of reported chronic disease in the sample seems low.  Are these numbers representative of their respective countries?  Is there any evidence that the sample was healthier than these countries as a whole?

2. Were rural-urban differences considered in these countries?  Is there any evidence that rurality matters as in the US?

3.  Can an argument be made that the government should leave it to primary care physicians and other healthcare professionals to address the vaccine issue with citizens and that the government should not be involved?

4.  Did these country’s governments have the same level of public health information campaigns about the vaccine and were there attempts at invoking vaccine mandates in any of these countries?  I am wondering if some of the country differences are due to specific differences in how these governments handled the response to the pandemic.

5.  It is extremely interesting that 80% of the citizens of these Balkan nations embrace at least one conspiracy related to COVID-19, and this is not related to educational attainment. What was the relationship between health literacy and education.  Is there possibly an interaction between the two? They seem to be, mostly, moving in opposite directions in the association with societal trust.  This is curious. Can more be done to tease out why that is?

6.  Was the Societal Trust Scale normally distributed?  Were assumptions check before using it as a dependent variable in a linear regression model?  I am assuming it was, but it would be good to state so, since no transformation was applied.

Author Response

Dear reviewer,

Thank you very much for your time, insightful comments, recommendations and encouragement to revise and resubmit the manuscript. The manuscript has been revised according to your requirements and suggestions as follows.

On behalf of all the authors,

Vida Jeremic Stojkovic

Response to comments:

Q1: The frequency of reported chronic disease in the sample seems low.  Are these numbers representative of their respective countries?  Is there any evidence that the sample was healthier than these countries as a whole?

RE1: Thank you for this question. We believe that low frequency of reported chronic diseases can be attributed to the relatively younger age of our respindents, given that we included only “younger adults”, in the age period ranging from 18 to 45 years, based on Levinson’s theory of individual life structure.

Q2: Were rural-urban differences considered in these countries?  Is there any evidence that rurality matters as in the US?

RE2: Unfortunately, we didn’t have the question inquiring the type of settlement. Thank you for this suggestion, we will definitively include it in the future studies.

Q3: Can an argument be made that the government should leave it to primary care physicians and other healthcare professionals to address the vaccine issue with citizens and that the government should not be involved?

RE3: This is a good point. However, large majority of health-care practice in Western Balkans is state-funded, so it would be difficult to exclude the involvement of government form the people’s perception.

Q4: Did these country’s governments have the same level of public health information campaigns about the vaccine and were there attempts at invoking vaccine mandates in any of these countries?  I am wondering if some of the country differences are due to specific differences in how these governments handled the response to the pandemic.

RE4: At the time of conducting the survey, vaccination in all five studied countries was free of charge, provided by the state, and no mandates were imposed. Off course, there were differences in how each country organized vaccination-promotion campaign, but such analysis was beyond the scope of our study. However, it would be interesting to analyze vaccination-promotion campaigns in Western Balkans, thank you for this suggestion.

Q5: It is extremely interesting that 80% of the citizens of these Balkan nations embrace at least one conspiracy related to COVID-19, and this is not related to educational attainment. What was the relationship between health literacy and education.  Is there possibly an interaction between the two? They seem to be, mostly, moving in opposite directions in the association with societal trust.  This is curious. Can more be done to tease out why that is?

RE5: Thank you very much, this is very interesting observation. The relationship between health literacy and education also caught our attention, and we did analysis revealing that the coefficients are small, and, moreover, they were not statistically significant.

Q6: Was the Societal Trust Scale normally distributed?  Were assumptions check before using it as a dependent variable in a linear regression model?  I am assuming it was, but it would be good to state so, since no transformation was applied.

RE6:Guided by the rule that multiple analysis should include the variables that have shown significance in univariate analysis, multivariate linear regressions were performed for samples gained from Serbia, Bosnia and Herzegovina and North Macedonia (Table 4). Data satisfied conditions for multiple linear regression analysis. An inspection of the variance inflation factor (VIF) among the independent variables for the three samples did not reveal issues of multicollinearity (VIF ranged from 1.01 to 1.99).Given that health literacy has proven to be the only significant predictor in the samples collected in North Macedonia and Albania, the results of univariate analyses are presented.

Round 2

Reviewer 2 Report

The authors answered the questions I raised.